# Optimizing Mechanical and Electrical Performance of SWCNTs/Fe_3_O_4_ Epoxy Nanocomposites: The Role of Filler Concentration and Alignment

**DOI:** 10.3390/polym16182595

**Published:** 2024-09-13

**Authors:** Zulfiqar Ali, Saba Yaqoob, Alessandro Lo Schiavo, Alberto D’Amore

**Affiliations:** 1Dipartimento di Matematica e Fisica, Università degli Studi della Campania “Luigi Vanvitelli”, Viale Abramo Lincoln n. 5, 81100 Caserta, Italy; zulfiqar.ali@unicampania.it (Z.A.); saba.yaqoob@unicampania.it (S.Y.); 2Dipartimento di Ingegneria, Università degli Studi della Campania “Luigi Vanvitelli”, Via Roma 29, 81031 Aversa, Italy; alessandro.loschiavo@unicampania.it

**Keywords:** epoxy composites, single-walled carbon nanotubes (SWCNTs), Fe_3_O_4_ nanoparticles, magnetic alignment, mechanical properties, electrical conductivity

## Abstract

The demand for polymer composites with improved mechanical and electrical properties is crucial for advanced aerospace, electronics, and energy storage applications. Single-walled carbon nanotubes (SWCNTs) and iron oxide (Fe_3_O_4_) nanoparticles are key fillers that enhance these properties, yet challenges like orientation, uniform dispersion, and agglomeration must be addressed to realize their full potential. This study focuses on developing SWCNTs/Fe_3_O_4_ epoxy composites by keeping the SWCNT concentration constant at 0.03 Vol.% and varying with Fe_3_O_4_ concentrations at 0.1, 0.5, and 1 Vol.% for two different configurations: randomly orientated (R-) and magnetic field-assisted horizontally aligned (A-) SWCNTs/Fe_3_O_4_ epoxy composites, and investigates the effects of filler concentration, dispersion, and magnetic alignment on the mechanical and electrical properties. The research reveals that both composite configurations achieve an optimal mechanical performance at 0.5 Vol.% Fe_3_O_4_, while A- SWCNTs/Fe_3_O_4_ epoxy composites outperformed at all concentrations. However, at 1 Vol.% Fe_3_O_4_, mechanical properties decline due to nanoparticle agglomeration, which disrupts stress distribution. In contrast, electrical conductivity peaks at 1 Vol.% Fe_3_O_4_, indicating that the higher density of Fe_3_O_4_ nanoparticles enhances the conductive network despite the mechanical losses. This study highlights the need for precise control over filler content and alignment to optimize mechanical strength and electrical conductivity in SWCNTs/Fe_3_O_4_ epoxy nanocomposites.

## 1. Introduction

Polymer composites have emerged as a class of advanced materials with applications spanning various industries due to their superior mechanical, electrical, and thermal properties. The versatility of polymer composites stems from their ability to be tailored by incorporating different fillers, which can significantly enhance their performance characteristics [1,2]. One of the most promising fillers for polymer composites is carbon nanotubes (CNTs) [3], particularly single-walled carbon nanotubes (SWCNTs). SWCNTs are known for their extraordinary mechanical strength, electrical conductivity, and thermal stability, making them ideal candidates for reinforcing polymers [4]. Numerous studies have demonstrated SWCNTs’ effectiveness in enhancing polymer composites’ properties. For instance, Arash et al. [5] evaluated the elastic properties of the interfacial region in CNT-reinforced poly(methyl methacrylate) (PMMA) composites using molecular dynamics simulations and reported that increasing the aspect ratio of CNTs enhances Young’s modulus and yield strength of both the interfacial region and the overall composite. On the other hand, Burkov et al. [6] reported carbon fiber-reinforced polymers (CFRP) modified with SWCNT, showing that while tensile properties see little change, flexural strength and modulus improve by 22% and 16% with 0.3 wt.% SWCNTs. However, electrical conductivity significantly increases, particularly out-of-plane. Thus, the incorporation of SWCNTs in polymer matrices not only enhances the mechanical strength but also contributes to the electrical conductivity of the composites.

However, the dispersion of SWCNTs within the polymer matrix poses a significant challenge due to their tendency to agglomerate. This issue can be addressed by employing suitable secondary fillers that aid in the dispersion of SWCNTs. Fe_3_O_4_ nanoparticles are particularly effective due to their magnetic properties and compatibility with CNTs. Adding Fe_3_O_4_ nanoparticles to SWCNTs not only facilitates better dispersion within the polymer matrix but also provides the feasibility of magnetic alignment of CNTs [7]. When subjected to a directional magnetic field, Fe_3_O_4_ nanoparticles align, thus aligning the attached SWCNTs. This alignment can significantly enhance the anisotropic properties of the composites, as demonstrated by Ma et al. [8] that aligning CNTs with residual Ni particles in epoxy resin under a 0.4 T magnetic field significantly enhances fracture toughness (KIC), achieving up to a 51% increase with 3 wt.% aligned CNTs transverse to crack growth. The aligned CNT composites show anisotropic KIC and improved electrical conductivity, though the effectiveness of KIC enhancement decreases at higher CNT loadings. Wang et al. [9] reported CNTs decorated with magnetite (Fe_3_O_4_) nanoparticles via an in situ solvothermal method at 500 °C using ferrocene and CNTs, and it was observed that the composites showed good ferromagnetic properties at room temperature with a saturation magnetization of 32.5 emu/g and a coercivity of 110 Oe. Another study [10] presented enhanced mechanical properties by aligning Fe_3_O_4_/multi-walled carbon nanotubes (MWCNTs) in epoxy resin using a low magnetic field. Fe_3_O_4_ nanoparticles were attached to MWCNTs via a water-in-oil method, and the aligned modified MWCNTs in the resin show improved mechanical performance.

Recent advancements in the development of polymer nanocomposites have focused not only on chemical modifications but also on various physical modification techniques to enhance their mechanical, electrical, and thermal properties. For example, solvothermal and hydrothermal processing utilizes high temperatures and pressures to facilitate better integration of fillers. Cryogenic grinding and high-energy ball milling refine the dispersion of nanoparticles, resulting in improved composite performance. Layer-by-layer assembly offers precise control over filler distribution, while surface functionalization enhances nanoparticle compatibility with the polymer. Each method contributes to developing composites with enhanced characteristics, highlighting the importance of choosing the appropriate technique based on the specific requirements of the application. Shcherbakov et al. [11] investigated the use of microwave irradiation for its impact on unsaturated polyester composites modified with MWCNTs and demonstrated that microwave treatment at various stages of composite preparation significantly influenced the interaction between MWCNTs and the polyester matrix, enhancing the mechanical properties and thermal stability of the resulting composite due to improved dispersion and reduced agglomeration of the nanotubes. 

Similarly, the application of high-frequency currents has been explored to enhance the alignment and distribution of nanofillers in the polymer matrix. This technique can create localized heating effects that enhance the interfacial bonding between fillers and the matrix, leading to composites with superior mechanical and electrical properties. Choi et al. [12] explored the application of plasma treatment to boron nitride nanosheets (BNNS) in epoxy nanocomposites to enhance thermal conductivity and mechanical properties and reported that plasma treatment improved the dispersion of BNNS within the epoxy matrix, resulting in substantial enhancements in thermal conductivity and fracture toughness. The plasma treatment facilitated better interfacial adhesion and reduced nanoparticle aggregation, which contributed to the improved performance of the nanocomposites.

Several studies support the alignment of CNTs using magnetic fields. For instance, research has shown that strong magnetic fields can align CNTs at a macroscopic scale, forming long wires parallel to the magnetic field lines [13]. This alignment technique is advantageous for creating composites with enhanced directional properties. Additionally, modulated magnetic fields have been demonstrated to achieve simultaneous alignment and micropatterning of CNTs within polymer composites. This method leverages the magnetic anisotropy of CNTs to control their orientation and patterning within the composite matrix [14]. Recent advancements in the field have further explored the dynamic alignment of single-walled carbon nanotubes in pulsed magnetic fields, highlighting the potential for precise control over CNT orientation [15].

Moreover, several theoretical frameworks support the observed alignment of magnetic nanoparticles under an external magnetic field well. The principles of magnetostatics describe the magnetic force, F, on a magnetic dipole, m, in a magnetic field, B, as F=∇(m·B), indicating that a magnetic dipole in a non-uniform field experiences a force that aligns it along the field gradient [16]. For ferromagnetic materials, the Weiss theory of ferromagnetism provides a critical understanding. This theory introduces the concept of internal molecular fields, represented by the equation: $H_{eff}=H+\lambda M$, where H_eff_ is the effective magnetic field, H is the applied magnetic field, λ is the Weiss constant, and M is the magnetization. The spontaneous alignment of magnetic moments in ferromagnetic materials is attributed to these internal fields, which are significantly strong, causing a parallel alignment even in the absence of an external field [17]. Furthermore, Maxwell’s equations, specifically Gauss’s law for magnetism (∇⋅B=0) and the Ampère-Maxwell law, describe the behavior of the magnetic field, ensuring that the field lines guide the alignment of Fe_3_O_4_ particles and the attached SWCNTs [18]. When preparing A- SWCNTs/Fe_3_O_4_ epoxy composites under a magnetic field, the concept of magnetic anisotropy helps explain why Fe_3_O_4_ particles align in a specific direction. Magnetic anisotropy refers to the tendency of a material to prefer magnetization in certain directions, with the easy axis being the direction where this magnetization occurs most naturally. The anisotropy energy $E_{A}=K_{u}{Sin}^{2}\theta$ (with K_u_ being the anisotropy constant that measures how strongly the material prefers this direction and θ the angle between the magnetization direction and the easy axis i.e., the direction where this magnetization occurs most naturally) indicates that the energy is minimized when the magnetization aligns with the easy axis (where θ = 0). When a magnetic field is applied during the preparation of the composites, the Fe_3_O_4_ particles tend to align their magnetization along the field direction because it corresponds to the direction of the lowest anisotropy energy. This alignment occurs because the system naturally seeks to minimize its energy, leading to the aligned structure of the SWCNTs/Fe_3_O_4_ composites within the epoxy matrix [19,20]. These studies underscore the importance of magnetic alignment techniques in fabricating high-performance nanocomposites with tailored properties. The strategic alignment of SWCNTs and Fe_3_O_4_ nanoparticles within an epoxy matrix under magnetic fields opens numerous avenues for advanced material design. Aligned SWCNT composites are particularly promising in electronics, where directional electrical conductivity is crucial, and in structural materials, where directional mechanical properties can significantly enhance performance. This study effectively demonstrates the significant influence of magnetic fields on the alignment of SWCNTs and Fe_3_O_4_ nanoparticles within an epoxy composite, supported by theoretical principles and recent advancements in the field, highlighting the potential for developing high-performance anisotropic materials.

In this study, we prepared SWCNTs/Fe_3_O_4_ epoxy composites with varying volume fractions of Fe_3_O_4_ (0.1, 0.5, and 1 Vol.%) and a constant 0.03 Vol.% SWCNTs. Our objective was to evaluate the impact of Fe_3_O_4_ content and SWCNTs alignment on the mechanical and electrical properties of the composites. We employed a detailed processing methodology involving sonication, mechanical stirring, and magnetic alignment to ensure optimal dispersion and alignment of the fillers within the epoxy matrix. The prepared composites were subjected to rigorous testing to compare the properties of aligned and randomly oriented samples, thereby providing insights into the benefits of magnetic alignment in enhancing the performance of SWCNT-Fe_3_O_4_/epoxy composites. This research contributes to the growing knowledge of advanced polymer composites and their potential applications in various high-performance domains.

## 2. Materials and Methods

### 2.1. Materials

Single-walled carbon nanotubes (SWCNTs) were obtained from Tuball (Leudelange, Luxembourg), featuring an outer diameter of 1.6 ± 0.4 nm, a length of ≥5 μm, a surface area of 300 m^2^/g, and a carbon content of ≥80 wt.%. Iron oxide (Fe_3_O_4_) powder with a predominant size of 200 nm was sourced from INOXIA Ltd (Dunsfold Aerodrome, Surrey, UK). Pure acetone (≥99.9%) was acquired from KEMIPOL SRL (Scerne di Pineto, Teramo, Italy). IN2 epoxy resin (viscosity 325 mPa·s) and AT30 slow epoxy hardener (95–115 min pot-life) were purchased from Easy Composites (Stoke-on-Trent, Staffordshire, UK). Non-toxic silicon rubber liquid and hardener were obtained from Reschimica (Vigevano, Lombardy, Italy) for mold making.

### 2.2. Preparation of SWCNTs/Fe_3_O_4_ Epoxy Composites

In the preparation of single-walled carbon nanotube (SWCNTs)/Fe_3_O_4_ epoxy composites, a systematic approach was employed to investigate the effect of SWCNTs/Fe_3_O_4_ alignment on the mechanical and electrical properties of the composites. The volume fractions of Fe_3_O_4_ were varied at 0.1, 0.5, and 1 Vol.%, with a constant 0.03 Vol.% of SWCNTs in all samples. First, a precise amount of SWCNTs was dispersed in 100 mL of acetone. This dispersion was subjected to continuous sonication using a probe-sonicator for 30 min in an ice bath to avoid acetone evaporation and overheating of the sample. Sonication is critical in ensuring the SWCNTs are well-dispersed within the solvent, preventing agglomeration and enhancing their interaction with other components. Following the initial sonication, a calculated amount of Fe_3_O_4_ nanoparticles was added to the SWCNT-acetone solution. This mixture was then sonicated for an additional 30 min. This step aimed to ensure uniform dispersion of Fe_3_O_4_ particles, which possess excellent magnetic properties, within the SWCNT-acetone solution. Subsequently, the epoxy resin was gradually introduced into the solution under vigorous mechanical stirring at 1000 rpm. The stirring duration varied depending on the Fe_3_O_4_ content: 5 min for 0.1 Vol.%, 10 min for 0.5 Vol.%, and 15 min for 1 Vol.% Fe_3_O_4_. This step is crucial for thoroughly mixing the resin with the SWCNT/Fe_3_O_4_ dispersion, ensuring a homogeneous composite mixture. The composite solution was then placed in an oven at 80 °C for 5 h to evaporate the acetone, which is necessary to prevent any residual solvent from affecting the curing process and the properties of the final composite. After acetone removal, the mixture was subjected to a vacuum for 3 h to eliminate any entrapped air bubbles, which could otherwise compromise the structural integrity and performance of the composites. Once the acetone and air bubbles were removed, an appropriate amount of hardener was added to the composite mixture. This mixture was again placed under vacuum for 1 h to ensure no air bubbles were present. Subsequently, the composite was cast into silicon molds to form dumbbell-shaped samples suitable for mechanical and electrical testing. Three samples from each aspect ratio were cured in the presence of metal-alloy of neodymium, iron, and boron (NdFeB) permanent magnets, which are known for their strong magnetic field. The magnets had a surface field strength of approximately 1 Tesla, in rectangular shape (block magnets/rectangular magnets) with dimensions of 50 × 25 × 10 mm^3^ (length × width × height) to explore the influence of SWCNT/Fe_3_O_4_ alignment on composite properties, while the other three samples were cured without it. Later, these samples were post-cured in a heating oven at 100 °C for 5 h. Figure 1 represents the complete procedure of this preparation. Magnets were placed on a metallic surface for the aligned samples to create a directional magnetic field, exploiting the magnetic properties of Fe_3_O_4_ to align the SWCNTs along the field direction. This alignment process is critical for enhancing the anisotropic properties of the composites. After curing, both aligned (A-) and randomly (R-) oriented SWCNT/Fe_3_O_4_ epoxy composites were subjected to a series of tests to evaluate their mechanical and electrical properties. The aim was to assess how the alignment of SWCNTs/Fe_3_O_4_, facilitated by the magnetic properties of Fe_3_O_4_, influences the overall performance of the composites.

### 2.3. Characterization

A Zeiss AxioVert.A1 optical microscope (Oberkochen, Baden-Württemberg, Germany) was employed to observe the dispersion quality of the composite solution. The samples were prepared according to testing standard ASTM D638 [21] to ensure consistency in the specimen geometry. The mechanical properties of the composite were evaluated using a Zwick-Roell Z010 universal testing machine (Ulm, Baden-Württemberg, Germany) at room temperature, equipped with a 10 kN load cell, a 50 mm gauge length, and a crosshead speed of 5 mm/min. The resulting stress-strain curves were utilized to calculate the elastic modulus, E, ultimate tensile strength, UTS, and fracture strain %. Additionally, the electrical properties were analyzed using a Keithley DMM6500 6½-Digit Touchscreen Multimeter (Solon, OH, USA) with the four-probe method, which allowed us to measure contact resistance and subsequently calculate the sheet resistance (R_s_) using the sheet resistance equation, $R_{s}=4.53236\frac{\Delta V}{I}$, where R_s_ is the sheet resistance, ΔV is the change in voltage measured between the inner probes, and I is the current applied between the outer probes. If the thickness of the measured material is known, then the R_s_ can be used to calculate its electrical resistivity: $\rho =R_{s}\times t$ and finally electrical conductivity $\sigma =\frac{1}{\rho }$ in S/m.

## 3. Results and Discussion

### 3.1. Dispersion Assessment

In this study, SWCNTs/Fe_3_O_4_ epoxy composite solutions were prepared using sonication and stirring processes to ensure proper mixing of the constituents prepared with a constant concentration of SWCNT at 0.03 Vol.% while varying the Fe_3_O_4_ content at 0.1 Vol.%, 0.5 Vol.%, and 1 Vol.%. Meanwhile, these prepared solutions were analyzed under an optical microscope to assess the homogeneity and dispersion of the components.

Figure 2a and Figure 2b represent the dispersion quality of the composite with 0.1 Vol.% Fe_3_O_4_ content observed at 500 μm and 100 μm, respectively. In Figure 2a, at a larger scale (500 μm), the dispersion appears quite good, indicating that the Fe_3_O_4_ nanoparticles are well distributed throughout the epoxy matrix. However, upon closer inspection at 100 μm in Figure 2b, it becomes evident that while the Fe_3_O_4_ nanoparticles are indeed well dispersed, they are not sufficiently interacting with the SWCNTs to form a connected network structure. The red arrows in Figure 2b highlight the presence of Fe_3_O_4_ nanoparticles, highlighting their distribution. Despite the excellent dispersion, the low concentration of Fe_3_O_4_ does not facilitate enough interaction with SWCNTs to enhance the composite’s performance significantly. On the other hand, Figure 2c and Figure 2d illustrate the dispersion quality at 0.5 Vol.% Fe_3_O_4_, observed at 500 μm and 100 μm, respectively. At 500 μm in Figure 2c, the composite shows a good dispersion of Fe_3_O_4_ nanoparticles similar to the 0.1 Vol.% sample, but with a denser distribution. A closer look in Figure 2d, observed at 100 μm, shows the interaction between Fe_3_O_4_ nanoparticles and SWCNTs. The red arrows indicate regions where Fe_3_O_4_ nanoparticles act as connecting points, forming connected structures with the SWCNTs. This intermediate concentration allows for better interaction between the components, leading to fewer agglomerations and a more homogeneous composite. The improved dispersion and interaction suggest an optimal balance for enhancing the composite’s properties at this concentration. However, Figure 2e and Figure 2f depict the dispersion at the highest Fe_3_O_4_ content of 1 Vol.%, observed at 500 μm and 100 μm, respectively. At the larger scale (Figure 2e), the dispersion appears dense, indicating a high concentration of Fe_3_O_4_ nanoparticles. However, closer inspection at 100 μm in Figure 2f reveals significant agglomeration. The red arrows point to regions where Fe_3_O_4_ nanoparticle clusters, forming large agglomerates. These clusters hinder the dispersion quality and negatively impact the composite’s performance. The high concentration of Fe_3_O_4_ leads to excessive particle-particle interactions rather than interactions with SWCNTs, causing these undesirable clusters.

### 3.2. Magnetically Induced Alignment of SWCNTs/Fe_3_O_4_ Epoxy Composites

The preparation of SWCNTs/Fe_3_O_4_ epoxy composites was systematically explored to examine the effects of magnetic fields on the alignment of these constituents during the curing process. As shown in Figure 3a, a homogeneous dispersion of SWCNTs and Fe_3_O_4_ nanoparticles in acetone was initially achieved. As evidenced by the response of the dispersed solution to an external magnet, the Fe_3_O_4_ particles demonstrated significant magnetic susceptibility, confirming their ferromagnetic nature. This behavior suggests the potential for controlling the spatial orientation of the Fe_3_O_4_ nanoparticles and the attached SWCNTs within the epoxy matrix through the application of an external magnetic field. The homogeneous solution was subsequently divided into two parts for curing: one cured in the absence of a magnetic field, resulting in a random orientation of the SWCNTs and Fe_3_O_4_ particles (Figure 3b), and the other cured in the presence of a magnetic field, leading to the alignment of Fe_3_O_4_ particles along the field lines and, consequently, the SWCNTs as well (Figure 3c). This alignment produced a composite with anisotropic properties, enhancing specific physical characteristics such as electrical conductivity and mechanical strength in the direction of alignment.

These composites were investigated via scanning electron microscopy (SEM) to understand the effects of filler concentration, dispersion, and magnetic alignment. For the random configuration at 0.1 Vol.% Fe_3_O_4_, as shown in Figure 4a, the SEM image reveals a sparse, disjointed network of SWCNTs. The Fe_3_O_4_ particles, at this low concentration, do not significantly interact with the SWCNTs, leading to a minimal connection between the two materials. Since fewer Fe_3_O_4_ particles are present, their magnetic attraction to one another is weak, and they cannot form large, visible networks with the SWCNTs. This results in a disorganized and sparse appearance, where few SWCNTs are involved in network formation. The magnetic dipole–dipole interaction between the Fe_3_O_4_ particles is insufficient to cause significant clustering, leaving the SWCNTs dispersed without any strong association with the Fe_3_O_4_ particles.

In contrast, when a magnetic field is applied, as seen in Figure 4b for the aligned configuration at 0.1 Vol.% Fe_3_O_4_, a slight improvement in alignment is visible. The external magnetic field directs the Fe_3_O_4_ particles along its field lines, which in turn aligns some of the SWCNTs as well. However, due to the low concentration of Fe_3_O_4_, only a small portion of SWCNTs are aligned. This supports the idea that at low concentrations, not enough Fe_3_O_4_ particles are available to fully surround and guide the SWCNTs into alignment, leaving most SWCNTs randomly oriented. The magnetic field can overcome the weak internal magnetic forces between the Fe_3_O_4_ particles, but its effect is limited by the low particle concentration, resulting in only partial alignment of the network.

At 0.5 Vol.% Fe_3_O_4_, a significant change occurs in the random configuration, as shown in Figure 4c. The SEM image reveals a much larger, denser network of randomly oriented SWCNTs and Fe_3_O_4_ nanoparticles. As the concentration of Fe_3_O_4_ increases, more nanoparticles are available to interact with the SWCNTs, allowing larger, more complex networks to form. The magnetic dipole–dipole interactions between Fe_3_O_4_ particles become stronger, leading to more pronounced agglomeration. This causes a greater number of SWCNTs to be pulled into the network, resulting in a larger, more interconnected structure. At this concentration, the Fe_3_O_4_ particles begin to significantly influence the dispersion of SWCNTs, forming larger, random networks.

In Figure 4d, which corresponds to the aligned configuration at 0.5 Vol.% Fe_3_O_4_, the effect of the external magnetic field is much more pronounced. The SEM image shows that a well-aligned network of SWCNTs and Fe_3_O_4_ nanoparticles has formed, with the SWCNTs aligned along the direction of the magnetic field. At this concentration, enough Fe_3_O_4_ nanoparticles are present to effectively surround and connect with the SWCNTs, allowing the external magnetic field to guide both materials into alignment. The increased number of Fe_3_O_4_ particles allows the magnetic field to overcome its internal magnetic attractions, leading to a well-organized, aligned network. This demonstrates the efficiency of magnetic field-assisted alignment at moderate concentrations, where the balance between particle availability and external magnetic forces results in optimal alignment.

At the highest concentration of 1 Vol.% Fe_3_O_4_, as shown in Figure 4e for the random configuration, the SEM image reveals an even larger, denser network with clear signs of agglomeration. At this concentration, the number of Fe_3_O_4_ particles is so high that their magnetic dipole–dipole interactions become the dominant force, leading to significant clustering and agglomeration. The SWCNTs become entangled in these dense networks, resulting in the formation of large, disordered structures. The high concentration of Fe_3_O_4_ increases the viscosity of the composite, further promoting agglomeration and reducing the uniform dispersion of SWCNTs. The random network appears denser and less organized due to the strong internal magnetic forces pulling the Fe_3_O_4_ particles and SWCNTs into larger, more complex structures.

In the aligned configuration at 1 Vol.% Fe_3_O_4_, as shown in Figure 4f, the SEM image shows clear signs of agglomeration, with fewer aligned SWCNTs compared to the 0.5 Vol.% cases. While more Fe_3_O_4_ particles are available to interact with the SWCNTs, the increased viscosity and close proximity of the particles hinder the effectiveness of the magnetic field. The internal magnetic forces between the Fe_3_O_4_ particles become stronger than the external magnetic field’s influence, leading to agglomeration and reduced alignment. This results in the formation of larger, disordered agglomerates, with fewer SWCNTs successfully aligned along the magnetic field lines. The high concentration of Fe_3_O_4_ leads to a competition between internal magnetic interactions and the external magnetic field, with the former dominating, causing the alignment to be less effective at this concentration.

The underlying behavior of the Fe_3_O_4_/SWCNT composites in both the random and aligned configurations can be explained by the magnetorheological effect and the theory of magnetic dipole–dipole interactions. In the absence of an external magnetic field, Fe_3_O_4_ particles primarily interact through their internal magnetic dipole moments, causing them to attract each other and form random networks. As the concentration of Fe_3_O_4_ increases, these internal magnetic forces become stronger, leading to larger agglomerates. When an external magnetic field is applied, the Fe_3_O_4_ particles experience a magnetic force that aligns them along the field lines, overriding their internal magnetic attractions. However, at high concentrations, the internal magnetic forces between Fe_3_O_4_ particles can become stronger than the external magnetic field, leading to agglomeration and reduced alignment [22,23].

This behavior can be quantified using the equation for the magnetic force $F_{mag}$ acting on a particle in a magnetic field gradient as follows:(1)$$F_{mag}=\frac{3}{2}\mu_{0}V\chi \frac{dH^{2}}{dx}$$
where, $\mu_{0}$ is the magnetic permeability of free space, V is the volume of the Fe_3_O_4_ particles, χ is the magnetic susceptibility of the particles, $\frac{dH^{2}}{dx}$ is the gradient of the magnetic field.

This equation explains how the Fe_3_O_4_ particles experience a force in the presence of a magnetic field, which guides their movement and alignment. At low concentrations, the external magnetic field can easily align the Fe_3_O_4_ particles, leading to aligned SWCNT networks. However, at higher concentrations, the internal magnetic forces between Fe_3_O_4_ particles become significant, leading to agglomeration and reduced alignment.

### 3.3. Tensile Testing

Due to the exceptional characteristics of the fillers, epoxy composites reinforced with SWCNTs and Fe_3_O_4_ nanoparticles exhibit significantly enhanced mechanical properties, including increased tensile strength, modulus, toughness, and hardness. With their high tensile strength and modulus, SWCNTs significantly improve load transfer and resistance to crack propagation, Fe_3_O_4_ nanoparticles add rigidity and create additional barriers to crack growth, contributing to overall toughness and hardness. The orientation of these fillers within the epoxy matrix is crucial; aligned SWCNTs provide superior mechanical properties in the direction of alignment, leading to anisotropic behavior, while random orientations reduce these enhancements due to less effective load transfer and energy dissipation. Similarly, the orientation of Fe_3_O_4_ particles, particularly when aligned using magnetic fields, can influence the composite’s stiffness and strength anisotropically. When combined, the fillers’ orientations offer tailored mechanical properties depending on application needs. However, random orientation generally decreases the mechanical advantages of these fillers, making the composite less effective than when the fillers are strategically aligned [24,25]. Figure 5 depicts a tensile testing setup using a Zwick-Roell universal testing machine. The setup consists of a computer connected to the machine for data acquisition and control. The highlighted area shows the critical components of the tensile test, including the jaws that securely hold the sample and apply traction and an extensometer that measures the deformation of the sample during the test. The sample, positioned between the jaws, is subjected to tensile forces, and the extensometer records the elongation, providing data to evaluate the material’s mechanical properties.

Figure 6a presents stress-strain curves for R- and A- SWCNTs/Fe_3_O_4_ epoxy composites compared with pure epoxy. Pure epoxy shows the lowest stress at any given strain, indicating the inherent limitations in its load-bearing capacity. Introducing SWCNTs and Fe_3_O_4_ into the epoxy matrix significantly enhances its mechanical properties due to the superior load transfer capability of the SWCNTs and the reinforcing effect of Fe_3_O_4_ nanoparticles.

Among the composites, the A-0.03 SWCNTs/0.5 Fe_3_O_4_ epoxy composite exhibits the highest stress values, suggesting the optimal synergy between SWCNT alignment and Fe_3_O_4_ content, which maximizes the load transfer efficiency and enhances the tensile strength. The enhanced performance of the A- SWCNTs/Fe_3_O_4_ epoxy composites compared to the R- SWCNTs/Fe_3_O_4_ epoxy composites can be attributed to the more efficient load transfer provided by the horizontally aligned SWCNTs/Fe_3_O_4_. In the aligned composites, a more coherent network is formed that better supports the load and transmits stress throughout the matrix, improving the overall mechanical response.

However, at higher Fe_3_O_4_ concentrations (1 Vol.%), a decline in stress is observed, likely due to nanoparticle agglomeration, which creates stress concentration points and impairs the uniform dispersion of the reinforcing phases, thus reducing the composite’s overall mechanical performance. Additionally, the reduction in strain at higher stress levels across the composites indicates the materials’ increasing brittleness, where the enhanced stiffness due to the reinforcement limits the composite’s ability to undergo plastic deformation, leading to earlier fracture. This behavior is more pronounced in the A- SWCNTs/Fe_3_O_4_ epoxy composites.

Figure 6b illustrates the comparison of elastic modulus of pure epoxy, R- SWCNTs/Fe_3_O_4_ epoxy composites, and A- SWCNTs/Fe_3_O_4_ epoxy composites, each with a fixed SWCNTs content of 0.03 Vol.% and varying Fe_3_O_4_ content of 0.1, 0.5, and 1 Vol.%. The results demonstrate that the elastic modulus increases with the addition of both R- SWCNTs/Fe_3_O_4_ and A- SWCNTs/Fe_3_O_4_ as compared to pure epoxy, indicating enhanced mechanical properties. Furthermore, A- SWCNTs/Fe_3_O_4_ epoxy composites consistently exhibit higher elastic modulus values than those R- SWCNTs/Fe_3_O_4_ epoxy composites at all Fe_3_O_4_ concentrations, suggesting that the alignment of SWCNTs/Fe_3_O_4_ under a magnetic field significantly improves the stiffness of the composite. Specifically, at 0.1 Vol.% Fe_3_O_4_, the elastic modulus for R- SWCNTs/Fe_3_O_4_ is 4.19 GPa, while for A- SWCNTs/Fe_3_O_4_ it is 4.74 GPa. At 0.5 Vol.% Fe_3_O_4_, the values are 6.18 GPa for R- SWCNTs/Fe_3_O_4_ and 6.86 GPa for A- SWCNTs/Fe_3_O_4_. At 1 Vol.% Fe_3_O_4_, the modulus slightly decreases for both composites but remains higher in the A-SWCNTs/Fe_3_O_4_ composite (6.55 GPa) compared to the R- SWCNTs/Fe_3_O_4_ composite (5.12 GPa). This trend suggests that increasing the Fe_3_O_4_ content up to 0.5 Vol.% improves the stiffness of the composite, but further increase to 1 Vol.% results in a slight reduction in the elastic modulus, possibly due to the agglomeration of nanoparticles, which might disrupt the continuity. The overall enhancement in elastic modulus, especially in A- SWCNTs/Fe_3_O_4_ composites, underscores the importance of nanotube alignment in maximizing mechanical reinforcement in such nanocomposites.

Figure 6c compares the ultimate tensile strength (UTS) of pure epoxy and R-, A- SWCNTs/Fe_3_O_4_ epoxy composites. Pure epoxy serves as a baseline with a UTS of 47.246 MPa. Upon incorporating 0.1 Vol.% Fe_3_O_4_ into the epoxy with R- SWCNTs, the UTS increases to 52.405 MPa, indicating a modest enhancement. The corresponding A- SWCNTs/Fe_3_O_4_ epoxy composite shows a more significant increase in UTS to 57.087 MPa, suggesting that the alignment of SWCNTs/Fe_3_O_4_ contributes to better load transfer and composite reinforcement. As the Fe_3_O_4_ content increases to 0.5 Vol.%, the UTS of the R- SWCNTs/Fe_3_O_4_ composite jumps to 66.439 MPa, while the A-SWCNTs/Fe_3_O_4_ composite reaches 79.378 MPa, highlighting a substantial improvement due to the combined effects of Fe_3_O_4_ and SWCNT alignment. At the highest Fe_3_O_4_ content (1 Vol.%), the UTS slightly decreases to 59.313 MPa for the R- SWCNTs/Fe_3_O_4_ composite, whereas the A- SWCNTs/Fe_3_O_4_ composite maintains a high UTS of 73.068 MPa. This trend suggests that while Fe_3_O_4_ content enhances UTS up to a certain threshold, the alignment of SWCNTs/Fe_3_O_4_ is crucial in maximizing the tensile strength, particularly at higher filler concentrations. Overall, both Fe_3_O_4_ content and SWCNT/Fe_3_O_4_ alignment significantly influence the tensile strength of the epoxy composites, with aligned SWCNTs/Fe_3_O_4_ providing superior reinforcement. Figure 6d shows the variations in fracture strain (%) of R- and A- SWCNTs/Fe_3_O_4_ epoxy composites at different Fe_3_O_4_ loadings in comparison to pure epoxy with a fracture strain of 3.498%. With the introduction of SWCNTs/Fe_3_O_4_, the fracture strain decreases across all composites, reducing the material’s ability to withstand deformation before breaking. For R- SWCNTs/Fe_3_O_4_ composites, the fracture strain decreases from 3.104% at 0.1 Vol.% Fe_3_O_4_ to 2.053% at 0.5 Vol.% and further to 2.171% at 1 Vol.%. Similarly, the fracture strains of A- SWCNTs/Fe_3_O_4_ composites decrease from 2.981% at 0.1 Vol.% to 1.624% at 0.5 Vol.% and 1.95% at 1 Vol.%. Notably, the A- SWCNTs/Fe_3_O_4_ composites consistently exhibit lower fracture strain values compared to the R- SWCNTs/Fe_3_O_4_ composites at the same Fe_3_O_4_ content, indicating that the alignment of SWCNTs/Fe_3_O_4_ under a magnetic field further reduces the material’s ductility. This trend suggests that while adding Fe_3_O_4_ and aligning SWCNTs may improve specific properties like stiffness or strength, it compromises the material’s fracture strain, making it more brittle.

Figure 7a demonstrates the percentage (%) enhancement in the elastic modulus of SWCNTs/Fe_3_O_4_ epoxy composites, comparing two configurations: R- and A- SWCNTs/Fe_3_O_4_ epoxy composites. The percentage (%) enhancement for both composite configurations is calculated relative to pure epoxy using the following formula:(2)$$Elasticmodulus\left(\%\right)\;enhancement=\frac{E_{c}-E_{e}}{E_{e}}\%$$
where E_c_, is the elastic modulus of the SWCNTs/Fe_3_O_4_ epoxy composite and E_e_ is the elastic modulus of the pure epoxy.

The results show that A- SWCNTs/Fe_3_O_4_ epoxy composites consistently exhibit higher elastic modulus enhancements than R- SWCNTs/Fe_3_O_4_ epoxy composites at all Fe_3_O_4_ concentrations. Specifically, at 0.1 Vol.% Fe_3_O_4_, the R- SWCNTs composite shows a 64.96% enhancement, while the A- SWCNTs composite shows an 86.61% enhancement. At 0.5 Vol.% Fe_3_O_4_, the enhancements increase to 143.31% for R- SWCNTs and 170.08% for A- SWCNTs. At 1 Vol.% Fe_3_O_4_, they are 101.57% for R- SWCNTs and 157.87% for A- SWCNTs. These findings highlight that the alignment of SWCNTs/Fe_3_O_4_ significantly improves the elastic modulus, likely due to better load transfer and a more efficient reinforcing network. Both R- SWCNTs and A- SWCNTs composites exhibit substantial improvements in elastic modulus compared to pure epoxy, confirming the reinforcing effect of SWCNTs and Fe_3_O_4_, with alignment providing additional benefits.

Figure 7b presents the UTS enhancement (%) of SWCNTs/Fe_3_O_4_ epoxy composites, and the data show that at all Fe_3_O_4_ concentrations, the A- SWCNTs/Fe_3_O_4_ composites exhibit significantly higher UTS enhancement compared to the R- SWCNTs/Fe_3_O_4_ composites, indicating that the alignment of SWCNTs/Fe_3_O_4_ under a low magnetic field improves the tensile strength of the composites. Specifically, at 0.1 Vol.% Fe_3_O_4_, the UTS enhancement for A- SWCNTs is 20.83%, compared to 10.92% for R- SWCNTs. At 0.5 Vol.% Fe_3_O_4_, the A- SWCNTs show a substantial enhancement of 68.01%, whereas R- SWCNTs exhibit 40.64%. Finally, at 1 Vol.% Fe_3_O_4_, the UTS enhancement for A- SWCNTs is 54.67%, compared to 25.54% for R- SWCNTs. The trend clearly illustrates that the alignment of SWCNTs/Fe_3_O_4_ results in a more significant improvement in tensile strength across all Fe_3_O_4_ concentrations, with the maximum enhancement observed at 0.5 Vol.% Fe_3_O_4_ for A- SWCNTs. Figure 7c shows the fracture strain % decrement for SWCNTs/Fe_3_O_4_ epoxy composites using the following formula:(3)$$Fracturestrain\left(\%\right)\;decrement=\frac{\varepsilon_{e}–\varepsilon_{c}}{\varepsilon_{e}}\%$$
where $\varepsilon_{e}$, and $\varepsilon_{c}$ represent the fracture strains of pure epoxy and composite, respectively.

It is evident that as the Fe_3_O_4_ content increases from 0.1 to 1 Vol.%, there is a significant increase in the fracture strain decrement for both R- SWCNTs/Fe_3_O_4_ and A- SWCNTs/Fe_3_O_4_ epoxy composites. Specifically, at 0.1 Vol.% Fe_3_O_4_, the A- SWCNTs/Fe_3_O_4_ composite shows a fracture strain decrement of 14.45%, which is slightly higher than the 11.24% observed for the R- SWCNTs/Fe_3_O_4_ composite. As the Fe_3_O_4_ content increases to 0.5 Vol.%, the decrement rises significantly to 53.1% for the A- SWCNTs/Fe_3_O_4_ composite, compared to 41.32% for the R- SWCNTs/Fe_3_O_4_ composite. At 1 Vol.% Fe_3_O_4_, the fracture strain decrement reaches 44.2% for the A- SWCNTs/Fe_3_O_4_ composite and 37.9% for the R- SWCNTs/Fe_3_O_4_ composite. This trend suggests that the alignment of SWCNTs/Fe_3_O_4_ under a magnetic field leads to a more pronounced reduction in the material’s ductility.

Figure 7d presents % enhancement elastic modulus and UTS of A- SWCNTs/Fe_3_O_4_ epoxy composites as compared to R- SWCNTs/Fe_3_O_4_ epoxy composites. The graph reveals that A- SWCNTs composites presented a 10.49%, 4.25%, and 31.84% higher elastic modulus at 0.1, 0.5, and 1 Vol.% Fe_3_O_4_, respectively as compared to R- SWCNTs/Fe_3_O_4_ epoxy composites. On the other hand, the UTS of A- SWCNTs composites was 8.93%, 19.47%, and 23.19% higher than R- SWCNTs/Fe_3_O_4_ epoxy composites at 0.1, 0.5, and 1 Vol.% Fe_3_O_4_, respectively. This data underscores the superior mechanical performance of A- SWCNT composites. 

### 3.4. Electrical Conductivity Analysis

Pure epoxy typically exhibits very low electrical conductivity (10−9 S/m) due to its insulating nature, which arises from the absence of free charge carriers within its molecular structure. However, SWCNTs, known for their excellent electrical properties, form conductive pathways within the composite, while Fe_3_O_4_ nanoparticles enhance the overall network connectivity and potentially facilitate electron transfer. The alignment of SWCNTs under a magnetic field further optimizes these pathways, resulting in even higher conductivity than randomly oriented composites. Figure 8a illustrates the four-probe technique setup for measuring the electrical resistivity, *ρ*, to determine the electrical conductivity “σ” of SWCNTs/Fe_3_O_4_ epoxy composites where a digital multimeter (a Keithley model) is connected to the sample via four probes. These probes are aligned along the sample’s surface with equal spacing, s. Figure 8b outlines the circuit setup for the four-probe method. A constant current DC power supply injects current through the outer two probes (labeled 1 and 4), while the voltage drop is measured across the inner two probes (labeled 2 and 3). This configuration minimizes contact resistance effects, providing a more accurate resistance measurement.

Figure 9a presents the measured electrical conductivity for the R- and A- SWCNTs/Fe_3_O_4_ epoxy composites. The electrical conductivity increases with increasing Fe_3_O_4_ content for both types of composites, but the A- SWCNTs/Fe_3_O_4_ composites exhibit significantly higher conductivity than their random counterparts. For example, the conductivity for R- SWCNTs/Fe_3_O_4_ composites rises from 7.89 × 10^−6^ S/m at 0.1 Vol.% to 5.80 × 10^−5^ S/m at 1 Vol.%, while for the A- SWCNTs/Fe_3_O_4_ composites, it increases from 1.45 × 10^−5^ S/m to 7.35 × 10^−5^ S/m over the same range. This enhanced conductivity in A- SWCNTs/Fe_3_O_4_ composites is attributed to the better alignment of SWCNTs/Fe_3_O_4_ within the epoxy matrix, facilitated by the magnetic field, which promotes more effective charge carrier pathways. The increased alignment leads to a higher probability of electron transport along the conductive SWCNTs/Fe_3_O_4_ network, thereby boosting overall conductivity.

Figure 9b compares the % enhancement in electrical conductivity for both composites relative to pure epoxy, which serves as the baseline. The A- SWCNTs/Fe_3_O_4_ composites consistently demonstrate a higher enhancement across all Fe_3_O_4_ concentrations. For instance, at 1 Vol.% Fe_3_O_4_, the A- SWCNTs/Fe_3_O_4_ composite shows an enhancement of 7.35 × 10^9^%, whereas the R- SWCNTs/Fe_3_O_4_ composite achieves only 5.80 × 10^9^%. This dramatic difference underscores the critical role of SWCNT/Fe_3_O_4_ alignment in optimizing electrical properties. The magnetic alignment likely results in a more interconnected network of SWCNTs, which, combined with the conductive Fe_3_O_4_ nanoparticles, creates efficient pathways for electron flow. In contrast, the random orientation of SWCNTs in the R- SWCNTs/Fe_3_O_4_ composites leads to less efficient pathways, as the electrons must traverse more tortuous routes, reducing the overall conductivity enhancement. Therefore, applying a magnetic field during composite preparation is a significant factor in maximizing the electrical performance of these nanocomposites.

## 4. Conclusions

This study focuses on the impact of SWCNTs and Fe_3_O_4_ nanoparticles on the mechanical and electrical properties of epoxy composites, particularly examining two configurations: R- and A- SWCNTs/Fe_3_O_4_ epoxy composites. Both configurations were evaluated with a fixed SWCNT content of 0.03 Vol.% and varying Fe_3_O_4_ contents of 0.1, 0.5, and 1 Vol.%. Compared to the R- counterparts, the A- SWCNTs/Fe_3_O_4_ epoxy composites demonstrated superior performance across all Fe_3_O_4_ concentrations. Specifically, the A- composites exhibited significant improvements in elastic modulus, with increases of 86.61%, 170.08%, and 157.87% at 0.1, 0.5, and 1 Vol.% Fe_3_O_4_, respectively, whereas the R- composites showed lower enhancements of 64.96%, 143.31%, and 101.57% at the same Fe_3_O_4_ concentrations as compared to pure epoxy. The A- composites also achieved higher UTS with enhancements of 20.83%, 68.01%, and 54.57% compared to pure epoxy, while R- composites presented 10.92%, 40.64%, and 25.54% enhancement compared to pure epoxy at 0.1, 0.5, and 1 Vol.% Fe_3_O_4_, respectively. However, at 1 Vol.% Fe_3_O_4_, mechanical properties declined, likely due to nanoparticle agglomeration, which disrupts stress distribution. Despite this, electrical conductivity peaks at 1 Vol.% Fe_3_O_4_, as the higher filler concentration improves the conductive network, outweighing the mechanical drawbacks. In terms of electrical conductivity, the A- SWCNTs/Fe_3_O_4_ composites consistently outperformed the R- SWCNTs/Fe_3_O_4_ composites across all tested Fe_3_O_4_ concentrations, showing enhanced electrical conductivities of 1.45 × 10^9^% at 0.1 Vol.%, 5.25 × 10^9^% at 0.5 Vol.%, and 7.35 × 10^9^% at 1 Vol.% as compared to pure epoxy. The increased conductivity in the A- composites was due to the aligned SWCNTs forming more continuous conductive pathways, reducing electron scattering and improving overall electron transport. In contrast, the R- composites exhibited less efficient electron transport due to the disordered arrangement of SWCNTs, which resulted in higher electrical resistance. Thus, the alignment of SWCNTs in the A- composites provided superior mechanical reinforcement and electrical performance compared to the random orientation. This study underscores the delicate balance between filler concentration and alignment to optimize mechanical strength and electrical conductivity in SWCNT/Fe_3_O_4_ epoxy nanocomposites.

## Figures and Tables

**Figure 1 polymers-16-02595-f001:**
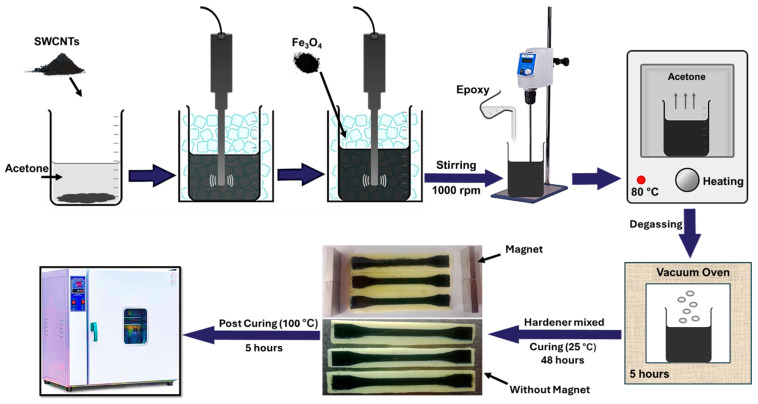
Schematic process for the preparation of SWCNTs/Fe_3_O_4_ epoxy composites and curing process between magnets and without magnets.

**Figure 2 polymers-16-02595-f002:**
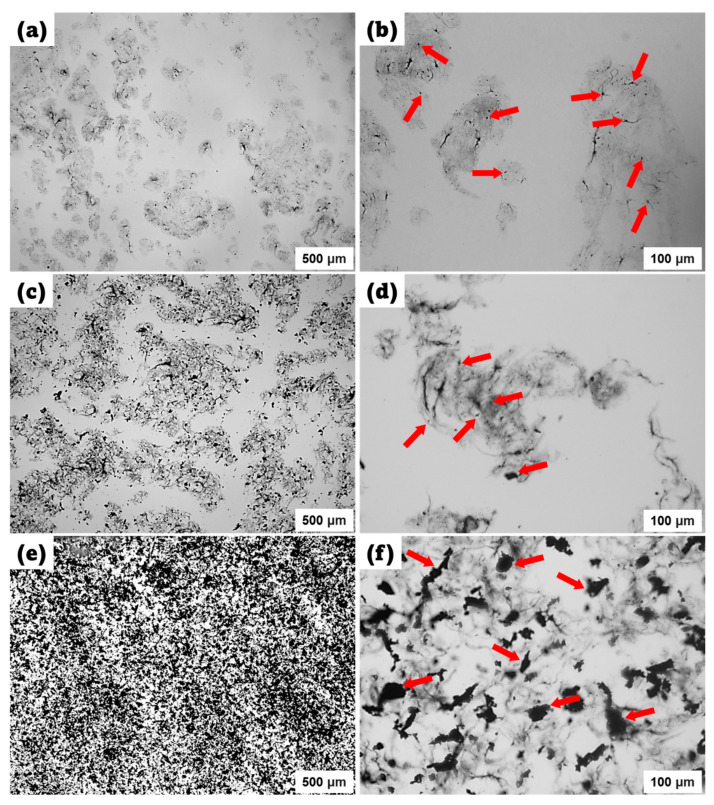
Microscopic observations for the dispersion analysis of SWCNTs/Fe_3_O_4_ epoxy composites. (**a**,**b**) 0.1 Vol.% Fe_3_O_4_: good dispersion, insufficient interaction (red arrows in (**b**)). (**c**,**d**) 0.5 Vol.% Fe_3_O_4_: good dispersion, better interaction (red arrows in (**d**)). (**e**,**f**) 1 Vol.% Fe_3_O_4_: poor dispersion, significant agglomeration (red arrows in (**f**)).

**Figure 3 polymers-16-02595-f003:**
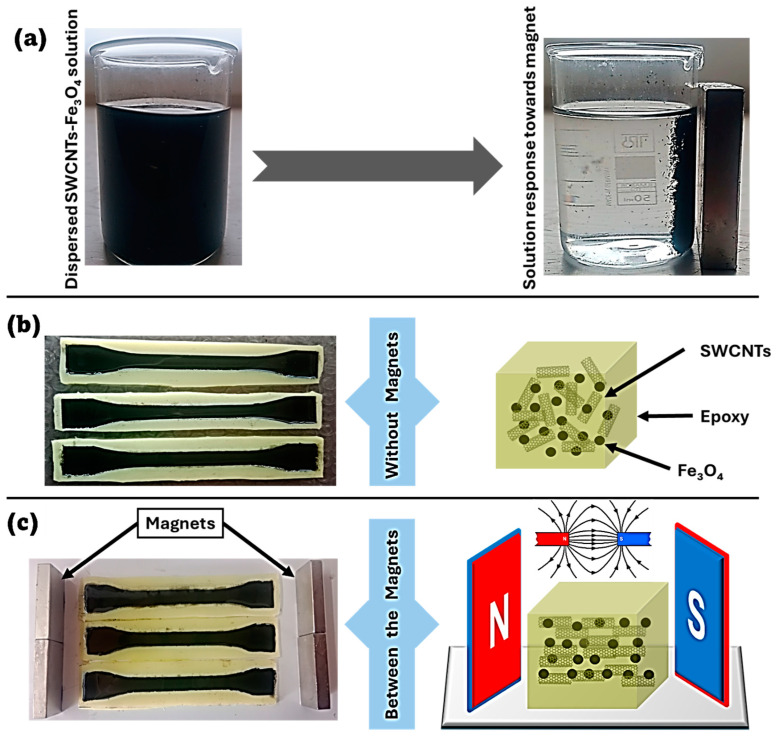
Representation of the orientation phenomenon in SWCNTs/Fe_3_O_4_ epoxy composites. (**a**) Dispersed SWCNTs/Fe_3_O_4_ solution and its magnetic response. (**b**) Randomly orientated (R-) SWCNTs/Fe_3_O_4_ epoxy composites. (**c**) Magnetic field-assisted horizontally aligned (A-) SWCNTs/Fe_3_O_4_ epoxy composites.

**Figure 4 polymers-16-02595-f004:**
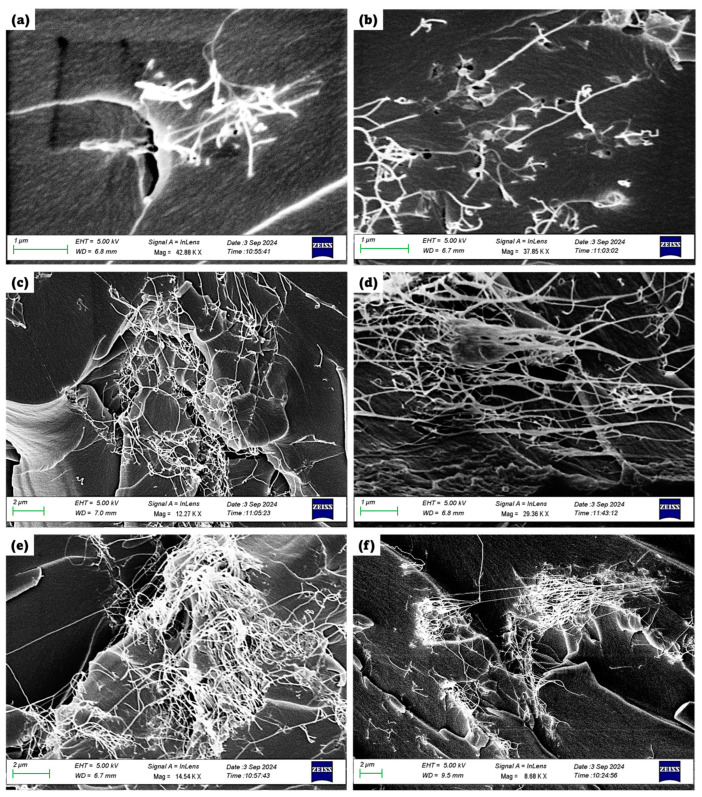
SEM images of R- and A- SWCNTs/Fe_3_O_4_ epoxy configurations at varying Fe_3_O_4_ concentrations. (**a**) R- SWCNTs/Fe_3_O_4_ epoxy at 0.1 Vol.% Fe_3_O_4_, (**b**) A- SWCNTs/Fe_3_O_4_ epoxy at 0.1 Vol.% Fe_3_O_4_, (**c**) R- SWCNTs/Fe_3_O_4_ epoxy at 0.5 Vol.% Fe_3_O_4_, (**d**) A- SWCNTs/Fe_3_O_4_ epoxy at 0.5 Vol.% Fe_3_O_4_, (**e**) R- SWCNTs/Fe_3_O_4_ epoxy at 1 Vol.% Fe_3_O_4_, and (**f**) A- SWCNTs/Fe_3_O_4_ epoxy at 1 Vol.% Fe_3_O_4_.

**Figure 5 polymers-16-02595-f005:**
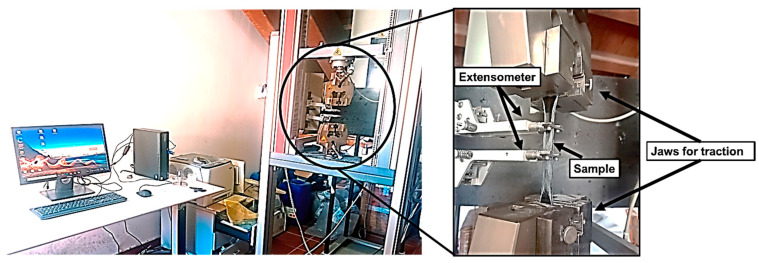
Tensile testing setup using Zwick-Roell universal testing machine with extensometer and sample in place.

**Figure 6 polymers-16-02595-f006:**
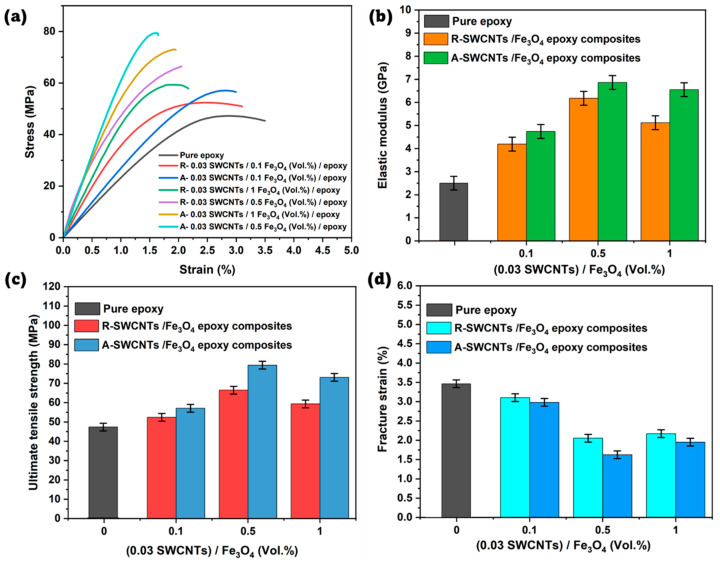
(**a**) Stress-strain curves for pure epoxy and R- and A- SWCNTs/Fe_3_O_4_ epoxy composites with 0.03 Vol.% SWCNTs and Fe_3_O_4_ concentrations of 0.1, 0.5, and 1 Vol. %. (**b**) elastic modulus (**c**) UTS and (**d**) fracture strain comparison.

**Figure 7 polymers-16-02595-f007:**
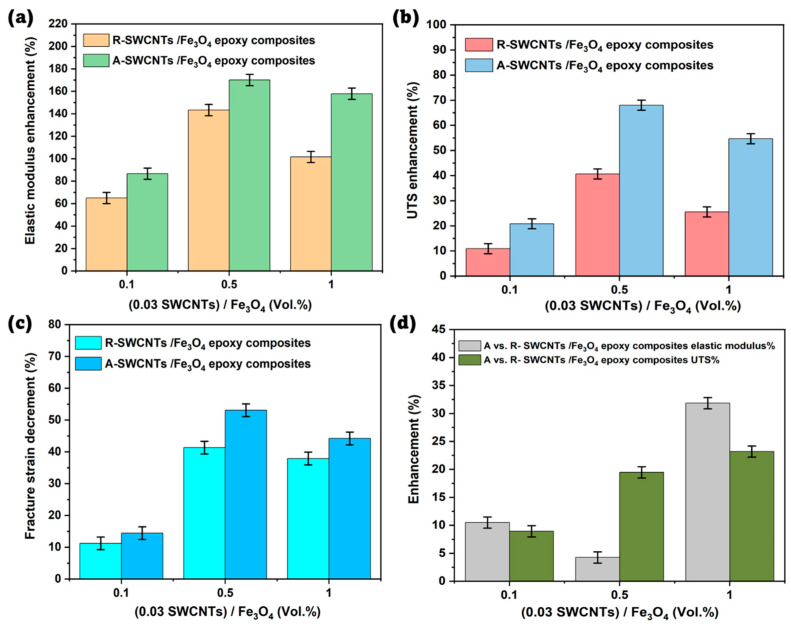
Comparison of R- and A- SWCNTs/Fe_3_O_4_ epoxy composites with increasing filler loading to pure epoxy for (**a**) elastic modulus enhancement (%), (**b**) UTS enhancement (%), (**c**) fracture strain decrement (%), and (**d**) elastic modulus and UTS enhancement (%) of A- SWCNTs/Fe_3_O_4_ epoxy composites compared to R- SWCNTs/Fe_3_O_4_ epoxy composites.

**Figure 8 polymers-16-02595-f008:**
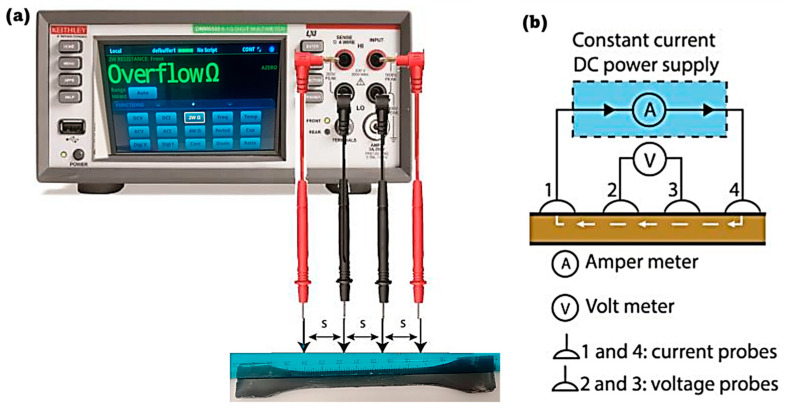
Schematic representation of the electrical conductivity measurement setup using the four-point probe method. (**a**) A digital multimeter connected to the sample for resistance measurement shows an overflow condition (because probes are not connected to the sample). (**b**) Schematic diagram illustrating the four-point probe configuration, where probes 1 and 4 are used for supplying current, and probes 2 and 3 are used for measuring voltage.

**Figure 9 polymers-16-02595-f009:**
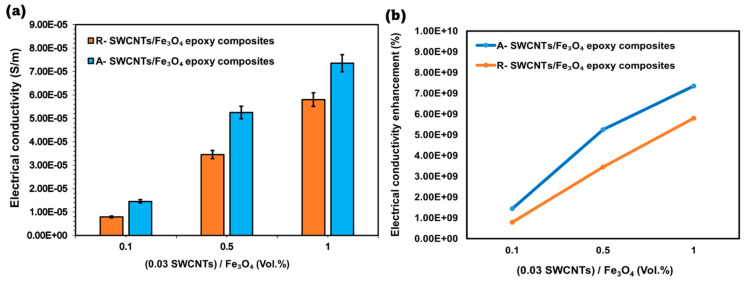
Electrical conductivity analysis of R- and A- SWCNTs/Fe_3_O_4_ epoxy composites at different volume fractions of Fe_3_O_4_. (**a**) Electrical conductivity (S/m) and (**b**) comparison of electrical conductivity enhancement (%) of R- and A- SWCNTs/Fe_3_O_4_ epoxy composites as compared to pure epoxy.

## Data Availability

Data is contained within the article.

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
