# Peer review of "Optimizing Mechanical and Electrical Performance of SWCNTs/Fe3O4 Epoxy Nanocomposites: The Role of Filler Concentration and Alignment"

_polymers, 2024, doi:10.3390/polym16182595_

Round 1
Reviewer 1 Report
Comments and Suggestions for Authors
The article entitled "Optimizing Mechanical and Electrical Performance of SWCNTs/Fe₃O₄ Epoxy Nanocomposites: The Role of Filler Concentration and Alignment" is written on the topical issue of targeted regulation of the functional properties of epoxy nanocomposites. The authors proposed and investigated an effective method for physical modification of the compositions, which consists in magnetic treatment of the epoxy composition containing nanoparticles. The studies were carried out using modern research methods and the results obtained are competently described. However, the reviewer has some comments that require correction before I can recommend this manuscript for publication:
1. Section 2.2. The characteristics of the magnets must be specified in this section.
2. Section 2.3. All calculation formulas must be specified in this section, and references to testing standards must be provided.
3. Why did not you study the different positions of the magnets relative to the samples for the properties of epoxy composites?
4. In Fig. 5. it is necessary to add data on the physical and mechanical properties of composites containing only 0.03 vol% SWCNTs and the results obtained must be compared with these values ​​to show the effectiveness of introducing Fe3O4 particles and magnetic treatment.
5. It is necessary to study the structure of the chips of the samples (SEM) for samples of type A and R.
6. In the "Introduction" section, it is necessary to consider the effect of various methods of physical modification (microwave, high-frequency current, etc.) on the properties of polymer nanocomposites. I recommend studying and citing the following articles:
- Polymers 2022, 14 (21), 4594; https://doi.org/10.3390/polym14214594
- Nanomaterials 2023, 13, 138. https://doi.org/10.3390/nano13010138
Author Response
Please, find the answers to the reviewer comments (1) in the attached file

Reviewer 2 Report
Comments and Suggestions for Authors
This study highlighted the need for precise control over reinforcement content and alignment to optimize mechanical strength and electrical conductivity in SWCNTs/Fe₃O₄ reinforced epoxy matrix nanocomposites. Herein, the SWCNT concentration was kept constant at 0.03 Vol.% while the loading fraction of Fe₃O₄ was varied from 0.1 to 1.0 Vol.% with two different configurations: (a) randomly orientated and (b) magnetic field-assisted horizontally aligned SWCNTs/Fe3O4 reinforcement in epoxy composites. Moreover, the effects of filler concentration, dispersion, and magnetic alignment on the mechanical and electrical properties of the composites were explored. It was found that both composite configurations achieved an optimal mechanical performance at 0.5 Vol.% Fe₃O₄, while magnetic field-assisted horizontally aligned SWCNTs/Fe3O4 epoxy composites outperformed at all concentrations. However, at 1 Vol.% Fe₃O₄, mechanical properties declined due to nanoparticle agglomeration, which disrupted stress distribution. In contrast, electrical conductivity peaked at 1 Vol.% Fe₃O₄, indicating that the higher density of Fe₃O₄ nanoparticles enhanced the conductive network despite the loss in mechanical performance.
It is an interesting manuscript showing the effect of the alignment of CNT-coated iron oxide particles, on the mechanical and functional properties of epoxy matrix composites. The results are promising though these seem more theoretically predicted than practically obtained. The major missing aspects in the manuscript are the evidence of the alignment of CNTs, evidence of the decoration pattern of CNTs on oxide particles or vice versa, force of attraction to coat CNTs with oxide particles, mismatching of filler loading in composites with the images shown, and formation of percolation network in composites, which enhances the electrical conductivity of insulating epoxy. The authors are requested to modify the manuscript in the light of above-mentioned points.
Concerns/Queries/Issues
SWCNTs were individually dispersed, or you had bundles of SWCNTs? How can you verify?
How did you accurately measure the 0.03 vol% of SWCNTs in epoxy matrix?
Do you understand that SWCNTs at 0.03 vol% can form a percolation network to make the composites electrically conductive?
If carbon content in SWCNTs is ≥80 wt.%, what is the remaining content?
Iron oxide (Fe₃Oâ‚„) powder had a predominant size of 200 nm while Single-walled carbon nanotubes (SWCNTs) had outer diameter of 1.6 ± 0.4 nm. Kindly inform, if SWCNTs were decorated on oxide particles or oxide particles were decorated on SWCNTs?
Why SWCNTs should be decorated on oxide particles? What was the force of attraction between SWCNTs and oxide particles? Did you introduce positive and negative charges on the surfaces of two fillers? Or something else? OR was it a simple mechanical mixing?
“Later, these samples were post-cured in a heating oven at 100°C for 5 hours.”
Magnetic field was applied after adding hardener in the epoxy. If yes, do you think that the hardening of epoxy might have affected the alignment process of SWCNTs?
Post-curing was performed in oven at 100 oC without magnets. If yes, do you think that alignment was retained after the removal of magnets?
“The aspect ratios Fe₃Oâ‚„ were varied at 0.1, 0.5, and 1 Vol.%, with a constant 0.03 Vol.% of SWCNTs in all samples.” How were the aspect ratios varied? Kindly explain.
Sonication was performed for 30 minutes after adding SWCNTs in acetone and subsequently for 30 minutes after adding oxide particles. How can you ensure that 30 minutes are enough to disperse SWCNTs in acetone and then oxide particles in acetone. Do you have any evidences at both stages about the uniform dispersion?
“The stirring duration varied depending on the Fe₃Oâ‚„ content: 5 minutes for 0.1 Vol.%, 10 minutes for 0.5 Vol.%, and 15 minutes for 1 Vol.% Fe₃Oâ‚„.” How did you choose the stirring times? Is there any evidence of the uniform dispersion of oxide particles after the chosen stirring times?
“A Zeiss AxioVert.A1 optical microscope was employed to observe the dispersion quality of the composite solution.” and “these prepared solutions were analyzed under an optical microscope to assess the homogeneity and dispersion of the components.” DO you think an optical microscope can identify nano fillers in composites? Please explain.
Comprehensively explain the characterization section by providing the “standards” used along with the testing parameters.
Are the images in Fig.2 from SEM or from optical microscope. Please verify.
“In Figure 2 (a), at a larger scale (500 μm), the dispersion appears quite good, indicating that the Fe₃Oâ‚„ nanoparticles are well distributed throughout the epoxy matrix.” Apparently, it is not a good dispersion. Comment please!
“However, upon closer inspection at 100 μm in Figure 2 (b), it becomes evident that while the Fe₃Oâ‚„ nanoparticles are indeed well dispersed, they are not sufficiently interacting with the SWCNTs to form a connected network structure.” Do you think that the interaction of SWCNTs with oxide particles will make a network? Please explain how?
“The red arrows in Figure 2 (b) highlight the presence of Fe₃Oâ‚„ nanoparticles, highlighting their distribution.” The size of oxide particles in 200 nm. Can you see this size particles in epoxy matrix? Please explain. You see agglomerates of the size of 100-500 micrometer only.
“Despite the excellent dispersion, the low concentration of Fe₃Oâ‚„ does not facilitate enough interaction with SWCNTs to enhance the composite's performance significantly.” By looking at the images in Fig.2 it seems that the percentage of fillers in 40-50 vol%. but your loading fraction of filler is maximum 1vol%. How do you justify the images?
Porosity can also be seen in the images of Fig.2. Please explain.
How the magnetic response in Fig.3a confirms the orientation as shown in Fig.3 c. Please explain.
Lines 199-245. Please include it in Introduction section.
Do you have composite images showing the orientation of SWCNTs? Please include in the manuscript.
Fig. 5 shows that loading fraction of oxide particles has a pronounced effect on mechanical properties than orientation of SWCNTs. Also explain this aspect in the manuscript.
“It is evident that as the Fe₃Oâ‚„ content increases from 0.1 to 1 Vol.%, there is a significant increase in the fracture strain decrement for both R- SWCNTs/Fe₃Oâ‚„ and A- SWCNTs/Fe₃Oâ‚„ epoxy composites.” You mean to say is that fractured strain decreased. Am I right, If yes, please simplify the sentence.
Images in Fig. 6 c and d. Simply show the decrease in fracture strain than increase in the decrement in the strain, which is confusing.
In literature, it is given that CNTs have flexible structure, and some reports say that fracture strain of composites increases after adding CNTs in brittle matrix such as epoxy. Comment, please!
Fig. 8. shows that both random and aligned CNTs in composites increase electrical conductivity but why this increase in electrical conductivity is due to the increase in the content of oxide particles? Because CNT content is same in all composites. Please explain.
Literature says that in CNT-polymeric matrix composites, electrical conductivity shoots up many orders of magnitude after the formation of percolation network. Yours composites are showing a linear rise. It means, percolation network in your composites was formed at 0.03% CNTs and 0.1% oxide particles, which is quite surprising. Comment, please!
Comments on the Quality of English LanguageMinor editing of English language is required.
Author Response
Please, find the answers to the reviewer comments (2) in the attached file

Round 2
Reviewer 1 Report
Comments and Suggestions for Authors
The authors made all necessary corrections to the manuscript and competently answered all the reviewer's questions. In this form, I recommend the manuscript for publication.
Reviewer 2 Report
Comments and Suggestions for Authors
Thank you for the revised manuscript. I am still unable to understand the following:
Are you sure these are SWCNTs in Fig.4 and not MWCNTs? Please look at their diameter!
Apparently, Fig.4 shows CNT-agglomerations and not aligned CNTs in the matrix. Look at your loading fraction: 0.03vol% and see the images. Do they match?
How can you observe SWCNTs in optical images?
SEM images showing the dispersed/aligned CNTs in matrix are missing.
Comments on the Quality of English LanguageMinor editing of English language required.